# Neuronal Network Modularity Determines Multi-Task Learning Capability

## Abstract

While highly modular biological neural networks excel at multi-domain cognitive processing, the computational principles underlying this evolutionary advantage remain unexplored. This study systematically quantifies how network modularity determines multi-task learning capabilities across diverse computational tasks. First, we analyzed connectome data and reveal their pronounced modular organization across all species. Further, to examine modularity's computational role, we designed a multi-task learning framework using structurally constrained recurrent neural networks trained on diverse data sets. Our key finding reveals a non-monotonic relationship between network modularity and multi-task learning performance. Performance degraded significantly at extreme modularity levels. Critically, single-task learning showed no systematic relationship with modularity, indicating that modular advantages are specific to scenarios requiring cross-task information flow. Moreover, we develop an information-theoretic framework that proves cross-module mutual information exhibits quadratic dependence on modularity. These findings provide quantitative insights into biological neural organization and offer design principles for artificial intelligence systems across multiple tasks.

Keywords: Network Neuroscience, Multi-Task Learning, Modularity, Connectome Analysis, Information Theory

## 1 Introduction

Biological neural systems exhibit remarkable modular organizational structures, with hierarchical functional differentiation spanning from microscopic neural circuits to macroscopic brain region specialization throughout the entire neural networkSporns & Betzel (2016); Bassett & Sporns (2017). For instance, in mammalian brains, distinct brain regions, such as the visual cortex, auditory cortex, and motor cortex possess highly specialized functions, with each module dedicated to processing specific types of information and executing corresponding computational tasksFelleman & Van Essen (1991). At a more refined level, neurons within individual cortical columns form local processing units that achieve specific perceptual or cognitive functions through dense internal connectionsDouglas & Martin (2004). This modular architecture is highly conserved throughout evolutionary history, from the 302 neurons of C. elegans to the 86 billion neurons in humans, all exhibiting significant community structures and functional partitioningTowlson et al. (2013); Van Den Heuvel & Sporns (2011). Remarkably, these modular networks demonstrate exceptional multi-domain cognitive processing capabilities, simultaneously handling diverse recognition tasks ranging from sensory pattern detection to complex behavioral controlSporns & Betzel (2016); Bassett & Sporns (2017).

To understand the functional significance of this ubiquitous modular organization, scientists have proposed multiple theoretical frameworks from diverse perspectives, ranging from evolutionary optimization to net-

work economy principles. The evolutionary adaptation hypothesis posits that modularity is an inevitable product of evolution. Kashtan and Alon demonstrated through theoretical models that when environmental tasks possess modular structures, evolution spontaneously generates modular network architecturesKashtan & Alon (2005). Wagner and colleagues further argued that modularity enables local optimization by reducing interference between different functions, thereby accelerating evolutionary adaptationWagner et al. (2007). Clune et al. discovered through computer simulations that under changing environmental pressures, modular networks exhibit greater evolutionary plasticity and adaptive capacity compared to non-modular networksClune et al. (2013). The network economy principle explains modularity from the perspective of resource optimization. Bullmore and Sporns proposed that brain networks exhibit a trade-off between minimizing wiring costs and maximizing information transmission efficiencyBullmore & Sporns (2012). Modular structures significantly reduce metabolic costs by increasing local connections while decreasing long-range connections. Computational studies by Kaiser and Hilgetag demonstrated that modularity can reduce total wiring length by up to nearly 50%Kaiser & Hilgetag (2006). Raj and Chen further proved that the brain's connectivity structure possesses near-optimal wiring costsRaj & Chen (2011). On the contrary, proponents of integrated information theory have proposed a contrary perspective. Tononi and colleagues argue that consciousness generation requires large-scale information integration, and excessive modularity would impede such integrationTononi et al. (1994). Furthermore, Shine et al. discovered through fMRI studies that when executing complex cognitive tasks, the brain dynamically reorganizes its modular structure, temporarily increasing connections between different modules to facilitate information integrationShine et al. (2016). Recent advances in computational neuroscience have provided new perspectives on this debate. Yang et al.Yang et al. (2019) observed that when training recurrent neural networks to perform multiple cognitive tasks, the networks automatically formed task-specific subnetworks, resembling functional parcellation in the brain. Duncker et al.Duncker et al. (2020) designed a learning rule aimed at minimizing interference between sequential learning tasks in recurrent networks, which can reuse similar dynamical structures across related tasks. This possibility of shared computation allows for faster learning during sequential training. Driscoll et al.Driscoll et al. (2024) further investigated compositional computational mechanisms in multitask learning, discovering that recurrent neural networks achieve flexible multitask computation through shared dynamical motifs. They demonstrated that networks can reuse fundamental computational modules to construct more complex behaviors.

However, despite the ubiquitous presence and evolutionary conservation of modular organization in neural systems, there is an urgent need to understand the computational principles that make modular architectures evolutionarily advantageous. To ground our investigation in biological reality, we first analyze and quantify the significant community properties of cross-species connectomes. Subsequently, we design a multi-task learning framework to train neural network backbones with varying degrees of network modularity. Finally, we propose a theoretical framework that elucidates how modularity influences multi-task learning performance.

## 2 METHODS

In this section, we will illustrate the multi-task learning method which we used to evaluate the influence of different modularity levels. For other details of metrics to evaluate the accuracy of the model, please refer to AppendixA.4.

### 2.1 MULTI TASK RNN

Based on the former studyLiu et al. (2016), we design a multi-task learning framework that leverages structurally constrained recurrent neural networks to investigate the relationship between network modularity and cross-task information flow. Our architecture systematically disentangles task-specific feature extraction from shared representational learning through a carefully designed modular structure.

### 2.1.1 NETWORK ARCHITECTURE DESIGN

Our framework adopts a hybrid architecture that systematically balances task specialization with shared representation learning. The model architecture flows from task-specific input encoders that mimic sensory-specific neural populations, through a shared recurrent backbone with biologically-constrained connectivity patterns, into an integration module analogous to association areas, and finally to task-specific output decoders representing specialized motor or cognitive outputs.

### 2.1.2 TASK-SPECIFIC INPUT ENCODING

Given $K$ distinct tasks with potentially heterogeneous input dimensionalities $\{d_1, d_2, \ldots, d_K\}$, we employ task-specific linear encoders to project each task's input space into a unified hidden representation space of dimension $H$. For task $k$ at time step $t$, the input encoder transforms the raw input $\mathbf{x}_t^{(k)}$ into an encoded representation $\mathbf{e}_t^{(k)}$:

$$\mathbf{e}_t^{(k)} = \mathbf{W}_{\text{enc}}^{(k)} \mathbf{x}_t^{(k)} + \mathbf{b}_{\text{enc}}^{(k)} \tag{1}$$

where $\mathbf{e}_t^{(k)} \in \mathbb{R}^H$ represents the encoded feature vector for task $k$ at time step $t$, $\mathbf{W}_{\text{enc}}^{(k)} \in \mathbb{R}^{H \times d_k}$ and $\mathbf{b}_{\text{enc}}^{(k)} \in \mathbb{R}^H$ are learnable parameters specific to task $k$. This design choice ensures that each task can learn optimal input transformations while feeding into a shared representational backbone, facilitating both task specialization and inter-task information flow.

### 2.1.3 BIOLOGICALLY CONSTRAINED RECURRENT BACKBONE

To systematically investigate how varying degrees of network modularity influence multi-task learning efficiency, we develop a connectome-constrained recurrent backbone that serves as a universal computational framework. The backbone consists of $L$ stacked recurrent layers, where each layer $\ell \in \{1, 2, \ldots, L\}$ implements biologically-inspired connectivity constraints through a binary adjacency matrix $\mathbf{A}^{(\ell)} \in \{0, 1\}^{H \times H}$ that determines the permissible synaptic connections within the neural population.

For layer $\ell$, the recurrent computation is formulated as:

$$\mathbf{i}_t^{(\ell)} = \mathbf{W}_{\text{ih}}^{(\ell)} \mathbf{h}_t^{(\ell-1)} + \mathbf{b}_{\text{ih}}^{(\ell)}, \tag{2}$$

$$\tilde{\mathbf{W}}_{\text{hh}}^{(\ell)} = \mathbf{W}_{\text{hh}}^{(\ell)} \odot \mathbf{A}^{(\ell)}, \tag{3}$$

$$\mathbf{r}_t^{(\ell)} = \tilde{\mathbf{W}}_{\text{hh}}^{(\ell)} \mathbf{h}_{t-1}^{(\ell)} + \mathbf{b}_{\text{hh}}^{(\ell)}, \tag{4}$$

$$\mathbf{h}_t^{(\ell)} = \tanh(\mathbf{i}_t^{(\ell)} + \mathbf{r}_t^{(\ell)}). \tag{5}$$

Where $\mathbf{W}_{\text{ih}}^{(\ell)} \in \mathbb{R}^{H \times H}$ and $\mathbf{W}_{\text{hh}}^{(\ell)} \in \mathbb{R}^{H \times H}$ are learnable weight matrices, $\mathbf{b}_{\text{ih}}^{(\ell)}, \mathbf{b}_{\text{hh}}^{(\ell)} \in \mathbb{R}^H$ are bias vectors, and $\odot$ denotes element-wise multiplication. The sparse weight matrix $\tilde{\mathbf{W}}_{\text{hh}}^{(\ell)}$ enforces structural constraints that define the network's modularity characteristics. $\mathbf{i}_t^{(\ell)} \in \mathbb{R}^H$ denotes the input-driven component at layer $\ell$ and time step $t$, representing the transformation of the current input (or the output from the previous layer) into the hidden state space. The term $\mathbf{r}_t^{(\ell)} \in \mathbb{R}^H$ corresponds to the recurrent component, capturing the influence of the previous hidden state $\mathbf{h}_{t-1}^{(\ell)}$ through synaptic connections governed by the sparse, modular structure encoded in $\mathbf{A}^{(\ell)}$. $\mathbf{h}_t^{(\ell)} \in \mathbb{R}^H$ represents the updated hidden state of layer $\ell$ at time $t$, obtained by passing the combined input and recurrent contributions through a nonlinear activation function (hyperbolic tangent), which models neuronal activation saturation.

### 2.1.4 TASK-SPECIFIC DECODING

The output layer comprises a shared two-layer feedforward transformation with dropout regularization and residual connections, followed by separate linear decoders for each task. The complete transformation pipeline is defined as:

$$\mathbf{z}_t^{(1)} = \text{ReLU}(\mathbf{W}_1 \mathbf{h}_t^{(L)} + \mathbf{b}_1), \tag{6}$$

$$\mathbf{z}_t^{(2)} = \text{Dropout}(\mathbf{z}_t^{(1)}, p), \tag{7}$$

$$\mathbf{z}_t^{(3)} = \text{ReLU}(\mathbf{W}_2 \mathbf{z}_t^{(2)} + \mathbf{b}_2), \tag{8}$$

$$\mathbf{f}_t = \mathbf{z}_t^{(3)} + \mathbf{h}_t^{(L)}, \tag{9}$$

$$\hat{\mathbf{y}}_t^{(k)} = \mathbf{W}_{\text{dec}}^{(k)} \mathbf{f}_t + \mathbf{b}_{\text{dec}}^{(k)}. \tag{10}$$

Where $\mathbf{W}_1, \mathbf{W}_2 \in \mathbb{R}^{H \times H}$, $\mathbf{b}_1, \mathbf{b}_2 \in \mathbb{R}^H$ are learnable parameters, $p$ is the dropout probability, and $\mathbf{W}_{\text{dec}}^{(k)} \in \mathbb{R}^{d_k^{\text{out}} \times H}$, $\mathbf{b}_{\text{dec}}^{(k)} \in \mathbb{R}^{d_k^{\text{out}}}$ are the task-specific decoding parameters for task $k$ with output dimension $d_k^{\text{out}}$. The residual connection preserves gradient flow while the task-specific decoders enable flexible output adaptation across heterogeneous tasks.

### 2.2 MODULARITY-CONTROLLED CONNECTIVITY PATTERNS

To systematically investigate the impact of network modularity on multi-task learning performance, we generate adjacency matrices $\mathbf{A}^{(\ell)}$ with controlled community structures. Each adjacency matrix is constructed using a community-based random graph model with $C$ communities.

The connectivity is governed by two parameters: the intra-community connection probability $p_{\text{in}}$ and the inter-community connection probability $p_{\text{out}}$. The modularity strength is controlled by varying the ratio $p_{\text{in}}/p_{\text{out}}$ while maintaining a fixed edge density $\rho = |E|/(H(H-1)/2)$ across all configurations.

The resulting modularity $Q$ is computed using the Newman-Girvan measureNewman & Girvan (2004):

$$Q = \frac{1}{2|E|} \sum_{i,j} \left[ \mathbf{A}_{ij} - \frac{k_i k_j}{2|E|} \right] \delta(c_i, c_j). \tag{11}$$

Where $k_i$ is the degree of node $i$, $c_i$ denotes the community assignment of node $i$, and $\delta(\cdot, \cdot)$ is the Kronecker delta function. For other properties of $Q$, please refer to Appendix. A.3.

## 3 EXPERIMENTAL RESULTS

Cross-species connectome (for connectome data description, please refer to Appendix A.1) analysis reveals pronounced modular organization across neural networks across four distinct species. Notably, while connection probability exhibits an inverse relationship with network size—decreasing from 0.0742 in *Ciona* tadpole to 0.000168 in *Drosophila* adult—all connectomes demonstrate significantly elevated modularity In addition, the modularity coefficient increases systematically across species, from 0.254 in *C. elegans* (1.57-fold higher than ER baseline) to 0.679 in adult *Drosophila* (5.94-fold enrichment). For other analysis, please read Zhang et al. (2025).

We further examine the relationship between Network modularity $Q$ and multi-task accuracy. The description of data sets and experimental settings are provided in Appendix A.2 and A.5, respectively. $Q$ exerts a substantial influence on multi-task learning performance. Across all tasks, the normalized MSE $\mu_{\text{norm}}$ (For

Table 1: Table of Network Statistics

| Connectomes | C. elegans | T. larva | D. larva | Drosophila |
|---|---|---|---|---|
| | | | | |
| **Network Size** | 302 Neurons
3557 Connections | 206 Neurons
3134 Connections | 2952 Neurons
191,980 Connections | 127,978 Neurons
2,613,129 Connections |
| **Connection Probability** | 0.0390 | 0.0742 | 0.0127 | 0.000168 |
| **Reciprocity** | 0.160 | 0.109 | 0.147 | 0.0790 |
| **Modularity** | 0.254
x1.57 than ER | 0.325
x2.30 than ER | 0.495
x4.91 than ER | 0.679
x5.94 than ER |

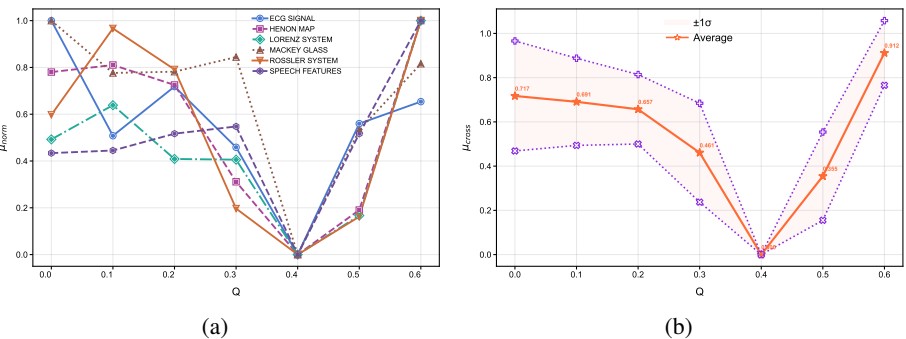

(a)  (b)

Figure 1: Change curves of normalized MSE $\mu_{norm}$ and mean normalized MSE $\mu_{\text{cross}}$ in multi-task learning experiments. (a) illustrates the variation curves of $\mu_{norm}$ for each task with respect to modularity $Q$. (b) shows the variation curves of $\mu_{\text{cross}}$ respect to $Q$.

metrics, please refer to AppendixA.4) exhibits a non-monotonic relationship with modularity, characterized by a sharp initial decline from $Q = 0.0$ to $Q = 0.40$, and culminating in a pronounced increase at $Q = 0.40$ to $Q = 0.60$ (Fig. 1(a)). The specific minimum varies across tasks, with some achieving optimal performance at medium-high modularity (e.g., Henon Map and Speech Features with $\mu_{\text{cross}} = 0$), while others show minima at different modularity levels.

In addition, the mean normalized MSE ($\mu_{\text{cross}}$) exhibits a pronounced minimum at $Q = 0.40$, demonstrating optimal performance at this modularity level. As modularity deviates from this optimal value, task performance degrades significantly in a non-monotonic manner. Specifically, when $Q$ decreases to the minimum, $\mu_{\text{cross}}$ increases dramatically to $0.717 \pm 0.249$, representing a significant deterioration. Similarly, when $Q$ increases to 0.60, the error rises to $0.912 \pm 0.146$.

Furthermore, to demonstrate that modularity variations solely impact multi-task learning, we individually trained each task using the original framework and calculated their respective normalized MSE $\mu_{\text{norm}}$. In stark contrast to the multi-task learning scenario, single-task learning exhibits no discernible pattern in the relationship between network modularity $Q$ and $\mu_{\text{norm}}$ (Fig. 2(a)). $\mu_{\text{norm}}$ display erratic and inconsistent trajectories across different modularity levels, with each task following its own idiosyncratic path. For instance, while Rossler System shows a sharp drop from $Q = 0.0$ to $0.10$, Henon Map demonstrates the opposite trend in the same range. In addition, some data sets (Lorenz System, ECG Signal, and Mackey Glass) reach their maximum error values at $Q = 0.40$, while other datasets maintain relatively low error.

To further examine the differential volatility patterns exhibited by multi-task learning and single-task learning across individual tasks under varying modularity conditions, we computed the coefficient of variation (CV) for each respective task. Stability analysis based on CV demonstrates that multi-task learning exhibits higher performance variability than single-task learning across all six tasks (Fig. 2(b)). Specifically, the CV values for multi-task learning range from 0.021 to 0.578, while single-task learning shows relatively stable CV values ranging from 0.001 to 0.402. The most significant difference occurs in the Henon Map task, where multi-task learning exhibits a coefficient of variation 0.399 higher than single-task learning, followed by the Rossler System task (difference: 0.353) and Speech Features task (difference: 0.313). For the sensitivity analysis, please refer to Appendix A.7.

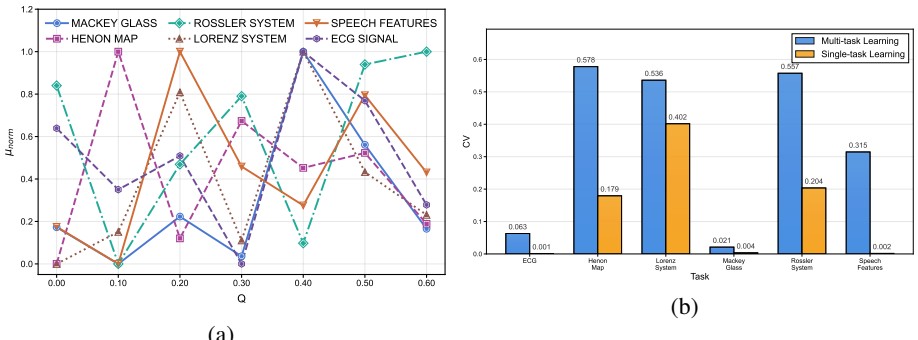

Figure 2: Change curves of normalized MSE $\mu_{norm}$ in singe-task learning experiments and the coefficient of variation $CV$ comparison between multi and single tasks. (a) illsutrate the variation curves of normalized MSE $\mu_{norm}$ for each task with respect to modularity $Q$ in single-task learning experiments. (b) shows the comparison of $CV$.

## 4 MODULARITY'S IMPACT ON INFORMATION FLOW

This section establishes a comprehensive theoretical framework to quantify how network modularity affects information transmission between modules. Building on foundational results from neural network theory Billingsley (2017); Lee et al. (2017), information theory Cover (1999), and network science Newman (2006), we derive novel bounds for cross-module information flow. Our main theoretical contribution proves that cross-module information flow exhibits a quadratic dependence on connection strength when the norm of connectivity matrix is sufficient small, providing new insight into why excessive modularity leads to degraded model performance through severely restricted inter-module communication.

### 4.1 MUTUAL INFORMATION AND CROSS-MODULE COVARIANCE ANALYSIS

To establish a tractable mathematical framework for analyzing information flow, we first assume that neural network activations asymptotically follow Gaussian distributions. This non-trivial result stems from the seminal work of Neal Billingsley (2017) and recent advances in neural network Gaussianity Lee et al. (2017); Matthews et al. (2018). Details can be referred at Appendix A.6. Actually, for a sufficient wide neural network with a large number of parameters, the output of each layer can reasonably be assumed to be approximately Gaussian. Neural Tangent Kernel (NTK) theory Jacot et al. (2018)Lee et al. (2019) further indicates that in the infinite-width limit, parameter updates remain confined to a small neighborhood, with changes diminishing to zero. In our model, the input dimension $H$ is set sufficient large, and training is limited to a small number of epochs. Thus, in the subsequent analysis, we adopt the Gaussian nature of activations, we now quantify how the strength of inter-module connections controls the statistical dependence between modules. Using standard results from control theory Anderson & Moore (2007) and matrix analysis Horn & Johnson (2012), we derive novel bounds that reveal how cross-module covariance scales with connection strength in modular architectures.

**Lemma 1.** *For jointly Gaussian random vectors $(H_i, H_j)$ representing activations from network modules $i$ and $j$, the mutual information can be expressed using standard information-theoretic formulas as:*

$$I(H_i; H_j) = -\frac{1}{2} \log \det(I - R_{ij}) \tag{12}$$

*where $R_{ij} = \Sigma_{ii}^{-1/2} \Sigma_{ij} \Sigma_{jj}^{-1} \Sigma_{ji} \Sigma_{ii}^{-1/2}$ is the normalized correlation matrix.*

*Proof.* Mutual information is defined by the standard formula Cover (1999):

$$I(H_i; H_j) = h(H_i) + h(H_j) - h(H_i, H_j) \tag{13}$$

For Gaussian distributions, differential entropy follows the standard expression Cover (1999):

$$h(X) = \frac{1}{2} \log \det(2\pi e \Sigma) \tag{14}$$

Therefore:

$$I(H_i; H_j) = \frac{1}{2} \log \frac{\det(\Sigma_{ii}) \det(\Sigma_{jj})}{\det(\Sigma)} \tag{15}$$

Applying the Schur complement decomposition Horn & Johnson (2012):

$$\det(\Sigma) = \det(\Sigma_{ii}) \det(\Sigma_{jj} - \Sigma_{ji} \Sigma_{ii}^{-1} \Sigma_{ij}) \tag{16}$$

Substituting yields:

$$I(H_i; H_j) = -\frac{1}{2} \log \det(I - \Sigma_{jj}^{-1} \Sigma_{ji} \Sigma_{ii}^{-1} \Sigma_{ij}) \tag{17}$$

Through similarity transformation and using the matrix identity $\det(I - UV^T) = \det(I - V^T U)$:

$$I(H_i; H_j) = -\frac{1}{2} \log \det(I - R_{ij}) \tag{18}$$

$\square$

**Theorem 1** (Cross-Module Covariance Bound)**.** *Consider a network with $M$ modules where the recurrent matrix $A$ has block structure, with $A_{ij}$ representing the connection matrix from module $j$ to module $i$. In steady state, the cross-module covariance satisfies:*

$$\|\Sigma_{ij}\| \leq c\|A_{ij}\| + O(\|A_{ij}\|^2), \tag{19}$$

*where $c$ is a constant depending only on intra-module connections.*

*Proof.* The steady-state covariance satisfies the discrete Lyapunov equationAnderson & Moore (2007):

$$\Sigma = A\Sigma A^T + Q. \tag{20}$$

Then, we perform perturbation analysis by setting cross-module connections as small quantities: $A_{ij} = \epsilon\tilde{A}_{ij}$ ($i \neq j$) where $\epsilon \ll 1$. Then we take the zero-order approximation ($\epsilon = 0$):

$$\Sigma_{ii}^{(0)} = A_{ii}\Sigma_{ii}^{(0)}A_{ii}^T + Q_{ii}, \quad \Sigma_{ij}^{(0)} = 0 \text{ for } i \neq j. \tag{21}$$

For the first-order correction, we retain first-order terms in $\epsilon$, and we find that cross-module covariance satisfies:

$$\Sigma_{ij} - A_{ii}\Sigma_{ij}A_{jj}^T = \epsilon(\tilde{A}_{ij}\Sigma_{jj}^{(0)}A_{jj}^T + A_{ii}\Sigma_{ii}^{(0)}\tilde{A}_{ij}^T) + Q_{ij}. \tag{22}$$

We recognize this as a Sylvester equation Horn & Johnson (2012) with solution:

$$\text{vec}(\Sigma_{ij}) = (I - A_{jj} \otimes A_{ii})^{-1}\text{vec}(\text{RHS}). \tag{23}$$

Since $(I - A_{jj} \otimes A_{ii})^{-1}$ is a bounded operator (by stability assumptions) and we observe that the right-hand side depends linearly on $\epsilon$, we obtain:

$$\|\Sigma_{ij}\| \leq c\|A_{ij}\| + O(\|A_{ij}\|^2). \tag{24}$$

$\square$

## 4.2 STRENGTH OF CONNECTIVITY MATRIX DETERMINES MUTUAL INFORMATION

This subsection establishes our central theoretical contribution: mutual information between modules exhibits quadratic dependence on connection strength. Building on standard results from information theory Cover (1999), we prove a novel scaling law that explains why highly modular networks suffer from information bottlenecks.

**Theorem 2** (Quadratic Scaling of Cross-Module Information Flow - *Main Theoretical Contribution*). *In highly modular networks where $\|A_{ij}\| \ll 1$ for all $i \neq j$, cross-module mutual information satisfies:*

$$I(H_i; H_j) = O(\|A_{ij}\|^2) \tag{25}$$

*This quadratic dependence explains the sharp performance degradation observed in over-modularized networks, as information flow becomes severely constrained when inter-module connections are weak.*

*Proof.* By Lemma 1, we shall analyze the scale of $\log\det(I - R_{ij})$. For small $\|R_{ij}\|$, we assume all elements of $R_{ij}$ share the same size, the dominant terms are:

$$\log\det(I - R_{ij}) \approx \log(1 - \text{tr}(R_{ij}) + O(\|R_{ij}\|^2)) \approx -\text{tr}(R_{ij}) + O(\|R_{ij}\|^2) \tag{26}$$

By our Theorem 1, $\|\Sigma_{ij}\| = O(\|A_{ij}\|)$, and we assume the size of $\|\Sigma_{ii}\|$, $\|\Sigma_{jj}\|$ to be some constant compared with $\|\Sigma_{ij}\|$, thus:

$$\|R_{ij}\| = \|\Sigma_{ii}^{-1/2}\Sigma_{ij}\Sigma_{jj}^{-1}\Sigma_{ji}\Sigma_{ii}^{-1/2}\| = O(\|\Sigma_{ij}\|^2) = O(\|A_{ij}\|^2) \tag{27}$$

Therefore:

$$I(H_i; H_j) = \frac{1}{2}\text{tr}(R_{ij}) + O(\|R_{ij}\|^2) = O(\|A_{ij}\|^2) \tag{28}$$

This establishes the novel quadratic scaling relationship between connection strength and information flow in modular networks. $\square$

Theorem 2 demonstrates that if the connection strength between two modules is too weak, the mutual information between the modules becomes minimal, making it difficult to form generalizable feature representations between tasks. Furthermore, our objective is to investigate the patterns of information transmission between modules when the connections among them are relatively strong. Based on the standard information theory Cover (1999), we have the following lemma.

**Lemma 2.** *Given a block-structured connectivity matrix $A$, where each module $A_{ij}$ is assumed to be of identical independently distributed, we derive the following bound on the cross-module mutual information:*

$$I(H_i; H_j) \geqslant H(A_{ij}), \tag{29}$$

*where $H_i$ and $H_j$ denote the activation outputs of blocks $i$ and $j$, respectively.*

*Proof.* Assume that the input variable is $\mathbf{X}$. Let $A_i$ and $A_j$ denote the corresponding row of block $i$ and $j$, we have

$$I(H_i, H_j) = I(A_i X, A_j X) = I(A_i, A_j | X) + h(X) \geqslant I(A_i, A_j | X). \tag{30}$$

The joint entropy of independent discrete random variables is indeed equal to the sum of their individual entropies. By the independence assumption, we have

$$
\begin{aligned}
I(A_i, A_j | X) = h(A_i) + h(A_j) - h(A_i, A_j) = H(A_{i1}, A_{i2}, \ldots, A_{iC}) + H(A_{j1}, A_{j2}, \ldots, A_{jC}) \\
- h(A_{i1}, \ldots, A_{iC}, A_{j1}, \ldots, A_{jC}) = h(A_{ij}).
\end{aligned} \tag{31}
$$

Due to the symmetric property of matrix $A$, $A_{ij}$ and $A_{ij}$ are in fact identical variables. Thus the term $h(A_{ij})$ is double-counted, leading to the final equality above, which completes the proof. $\square$

Generally, when we assume that the modularity is small, the connections between modules are usually non-negligible. If the connection strength between two modules is too strong, the dependency between the distributions of the two modules becomes excessive, resulting in a lack of module-specific information and leading to homogenization of the learned distributions.

## 5 CONCLUSION

Through comprehensive analysis of cross-species connectomes and systematic multi-task learning experiments, this study reveals that network modularity exhibits a non-monotonic relationship with multi-task learning performance, with optimal accuracy achieved at moderate modularity levels (Q=0.40). Our key findings demonstrate that while single-task learning shows no systematic relationship with modularity, multi-task scenarios exhibit significant performance degradation at extreme modularity levels.

Moreover, we establish a theoretical framework proving that cross-module mutual information exhibits quadratic dependence on connection strength when the norm of connectivity matrix is relatively small, providing mathematical foundations for understanding performance degradation in over-modularized networks. For high modularity, information exchange between modules becomes non-negligible or even excessive, which may result in a lack of module-specific information. The above theoretical contribution bridges information theory and network science to explain why biological neural systems evolved moderate modularity levels.

These findings offer critical insights for multiple disciplines. For neuroscience, our results explain the evolutionary conservation of modular brain architecture from C. elegans to humans, suggesting that moderate modularity optimizes cross-task information flow while preserving task-specific processing. For artificial neural networks, this research reveals biologically inspired design principles for multitask learning architectures, from neural networks to large language modelsRuder (2017). Our findings demonstrate that optimal performance requires balanced modularity to facilitate cross-task information flow while preserving high intra-modular connectivity for task-specific processingStandley et al. (2020). This principle could directly apply to large-scale foundation models to achieve efficient knowledge sharing across diverse tasks.

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

# A  APPENDIX

## A.1  CONNECTOME DATA SETS

The connectome is a complete network atlas that describes the connectivity relationships between all neurons in the brain or nervous systemSporns et al. (2005). We obtained connectome data from four species, including Caenorhabditis elegans (C. elegans), Drosophila larva, tadpole larva of Ciona, and adult Drosophila, for subsequent analysis.

The connectome of C. elegansCook et al. (2019) is the first organism in history to have its complete neural connectome mapped. This microscopic nematode possesses 302 neurons, and scientists utilized electron microscopy techniques to precisely delineate the synaptic connections between each neuron, establishing a comprehensive neural network atlas. This pioneering work laid the foundational framework for all subsequent connectomic studies and stands as a milestone achievement in the field of neuroscience.

The tadpole larva of Ciona constitutes the second animal to have its complete connectome mappedRyan et al. (2016). As a simple representative of chordates, the tunicate tadpole larva possesses a relatively simple yet complete nervous system that encompasses fundamental sensory, motor, and integrative functions. The mapping of this connectome has provided crucial insights into understanding the evolutionary origins and basic organizational principles of chordate nervous systems, serving as an important bridge linking invertebrate and vertebrate nervous system research.

A comprehensive study precisely delineated every neural connection within the Drosophila larval brainWinding et al. (2023), including key neural circuits responsible for learning, value computation, and action selection. This connectome encompasses thousands of neurons and tens of thousands of synaptic

connections, providing unprecedented detailed information for understanding the neural basis of insect intelligence and complex behaviors.

The adult Drosophila brain connectomeDorkenwald et al. (2024) represents one of the most complex complete brain connectivity maps currently available, containing over 120,000 neurons and more than 30 million synapses. This collaborative research effort by multiple international teams mapped the detailed connectome of the adult fly central brain, encompassing approximately 100,000 neurons and tens of millions of synaptic connections. This connectome not only elucidates the neural circuit foundations underlying complex Drosophila behaviors but also provides essential references for developing novel artificial intelligence algorithms and understanding higher-order animal brain functions.

## A.2 VALIDATION DATA SETS

We evaluate our approach on six diverse time series prediction tasks, comprising both synthetic chaotic systems and real-world applications. Each dataset presents unique challenges in terms of dynamics, dimensionality, and prediction complexity.

**Lorenz SystemLorenz (2017):** We generate the Lorenz attractor by numerically integrating the following differential equations:

$$\frac{dx}{dt} = \sigma(y - x) \tag{32}$$

$$\frac{dy}{dt} = x(\rho - z) - y \tag{33}$$

$$\frac{dz}{dt} = xy - \beta z \tag{34}$$

where $\sigma = 10.0$, $\rho = 28.0$, and $\beta = 8/3$. The system is integrated using the fourth-order Runge-Kutta method with a time step of $\Delta t = 0.01$ and initial conditions $(x_0, y_0, z_0) = (1.0, 1.0, 1.0)$.

**Hénon MapHénon (1976):** The Hénon map is a discrete-time dynamical system defined by:

$$x_{n+1} = 1 - ax_n^2 + y_n \tag{35}$$
$$y_{n+1} = bx_n \tag{36}$$

with parameters $a = 1.4$ and $b = 0.3$, initialized at $(x_0, y_0) = (0.0, 0.0)$.

**Rössler SystemRössler (1976):** The Rössler attractor is generated by integrating:

$$\frac{dx}{dt} = -y - z \tag{37}$$

$$\frac{dy}{dt} = x + ay \tag{38}$$

$$\frac{dz}{dt} = b + z(x - c) \tag{39}$$

with parameters $a = 0.2$, $b = 0.2$, $c = 5.7$, time step $\Delta t = 0.01$, and initial conditions $(x_0, y_0, z_0) = (1.0, 1.0, 1.0)$.

**Mackey-Glass Time SeriesMackey & Glass (1977):** We utilize the standard Mackey-Glass delayed differential equation dataset with delay parameter $\tau = 17$, following the preprocessing described in Mackey & Glass (1977).

**ECG Signal PredictionMcSharry et al. (2003):**We generate synthetic ECG signals following McSharry et al.'s approachMcSharry et al. (2003) with standard MIT-BIH parameters Moody & Mark (2001): 360 Hz

sampling rate, 75 bpm heart rate, including P-waves, QRS complexes, T-waves, baseline drift, and Gaussian noise ($\sigma = 0.05$).

**Speech FeaturesHammami & Bedda (2010):** We employ the Arabic Spoken Digits data set, consisting of 13-dimensional Mel-frequency cepstral coefficients (MFCCs) extracted from recordings of 10 Arabic digits (0-9) spoken by multiple speakers of both genders. The dataset contains 6,600 training examples (330 utterances $\times$ 2 genders $\times$ 10 digits).

### A.3 OTHER PROPERTIES OF MODULARITY

Define a $C \times C$ symmetric matrix $\mathbf{e}$ whose element $e_{ij}$ is the fraction of all edges in the network that link verticles in community $i$ to verticles in community $j$. The modularity can also be measured byNewman & Girvan (2004)

$$Q = \sum_i (e_{ii} - a_i^2), \tag{40}$$

where $a_i = \sum_j e_{ij}$, which present the fraction of edges that connect to vertices in community $i$. According to this expression, we can elucidate the range of modularity values.

**Proposition 1** (Range of modularity). *Given a matrix $W$ consisting of $C$ communities, the modularity satisfies*

$$0 \leqslant Q \leqslant \frac{C-1}{C} \tag{41}$$

*Proof.* For a certain community $C_k$, the intra-community connection probability is higher than that of random edge connection,

$$\sum_{c_i, c_j \in C_k} \mathbf{A}_{ij} \geqslant \sum_{c_i, c_j \in C_k} \frac{k_i k_j}{2|E|}. \tag{42}$$

For the left-hand side inequality, we have

$$Q = \frac{1}{2|E|} \sum_{i,j} \left[ \mathbf{A}_{ij} - \frac{k_i k_j}{2|E|} \right] \delta(c_i, c_j) = \frac{1}{2|E|} \sum_{k=1}^{C} \sum_{c_i, c_j \in C_k} \left[ \mathbf{A}_{ij} - \frac{k_i k_j}{2|E|} \right] \geqslant 0. \tag{43}$$

For the upper-bound estimation,

$$Q = \sum_{i=1}^{C} (e_{ii} - a_i^2) = \sum_{i=1}^{C} e_{ii} - \sum_{i=1}^{C} a_i^2 \tag{44}$$

$$\leqslant 1 - \sum_{i=1}^{C} a_i^2 \tag{45}$$

$$\leqslant 1 - \frac{(\sum_{i=1}^{C} a_i)^2}{C} = \frac{C-1}{C}. \tag{46}$$

The first inequality is due to $\sum_{i=1}^{C} e_{ii} \leqslant \sum_{i,j} e_{ij} = 1$, whereas the second inequality is obtained via the mean inequality. □

### A.4 EVALUATION METRICS

To systematically evaluate the performance of our models across different modularity levels and tasks, we employ a comprehensive evaluation framework which consists error measurement, variance analysis, normalization for cross-task comparison, and statistical aggregation across multiple tasks.

### A.4.1 MEAN SQUARED ERROR (MSE)

The Mean Squared Error serves as our primary performance metric for regression tasks, measuring the average squared differences between predicted and actual values:

$$\mu = \frac{1}{n} \sum_{i=1}^{n} (y_i - \hat{y}_i)^2, \tag{47}$$

where $n$ is the number of samples, $y_i$ represents the true values, and $\hat{y}_i$ denotes the predicted values. MSE provides several advantages for our evaluation: (1) it applies quadratic penalty to larger errors, making it sensitive to outliers; (2) smaller values indicate better model performance; and (3) it maintains the same units as the square of the original data.

### A.4.2 VARIANCE ANALYSIS

To complement our MSE analysis and understand the stability of model predictions, we compute the variance of the prediction errors:

$$\sigma = \frac{1}{n} \sum_{i=1}^{n} [(y_i - \hat{y}_i) - \mu]^2. \tag{48}$$

### A.4.3 NORMALIZED MSE VIA MIN-MAX SCALING

To enable meaningful comparison across tasks with different scales and units, we apply min-max normalization to the MSE values for each task independently:

$$\mu_{\text{norm}}^{(k)} = \frac{\mu^{(k)} - \min(\mu^{(k)})}{\max(\mu^{(k)}) - \min(\mu^{(k)})}, \tag{49}$$

where $\mu^{(k)}$ represents the MSE values across all modularity levels for task $k$. This normalization transforms all MSE values to the range $[0, 1]$, where 0 indicates the best performance (minimum MSE) and 1 represents the worst performance (maximum MSE) for each specific task.

### A.4.4 CROSS-TASK STATISTICAL AGGREGATION

To obtain overall performance statistics across all tasks, we compute the mean and standard deviation of the normalized MSE values:

$$\mu_{\text{cross}} = \frac{1}{K} \sum_{k=1}^{K} \mu_{\text{norm}}^{(k)}, \tag{50}$$

$$\sigma_{\text{cross}} = \sqrt{\frac{1}{K} \sum_{k=1}^{K} (\mu_{\text{norm}}^{(k)} - \mu_{\text{cross}})^2}, \tag{51}$$

where $K$ is the total number of tasks, $\mu_{\text{cross}}$ represents the cross-task mean performance, and $\sigma_{\text{cross}}$ denotes the cross-task standard deviation.

### A.4.5   COEFFICIENT OF VARIATION (CV)

To assess the relative variability of model performance across different tasks and modularity levels, we employ the coefficient of variation, which normalizes the standard deviation by the mean to provide a dimensionless measure of relative dispersion:

$$\text{CV} = \frac{\sigma_{\text{cross}}}{\mu_{\text{cross}}} \times 100\%. \tag{52}$$

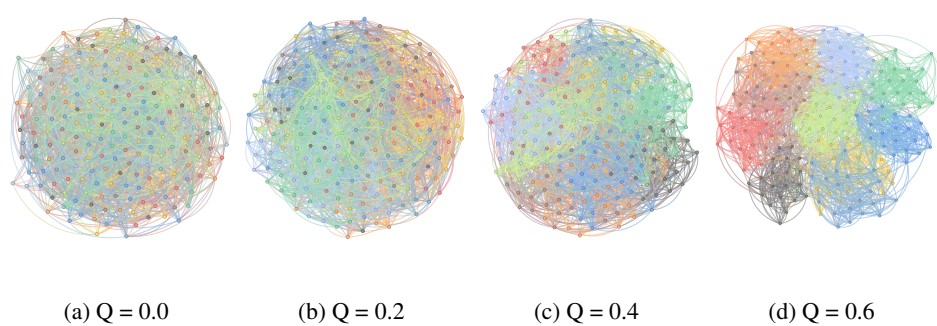

(a) Q = 0.0        (b) Q = 0.2        (c) Q = 0.4        (d) Q = 0.6

Figure 3: Visualization of graphs with different modularity $Q$.

### A.5   EXPERIMENTAL SETTINGS

We evaluate the impact of network modularity on multi and single task time series learning using a sparse RNN architecture with 200 hidden units across 3 layers and a fixed topology of 2000 edges. Seven network configurations are tested with modularity levels ranging from very high ($Q \approx 0.6$) to low ($Q \approx 0$) (Fig. 3), where each configuration maintains identical node and edge counts to isolate the effect of modular structure. Our evaluation encompasses six diverse time series prediction tasks, with each task treated as a sequence-to-sequence regression problem. The data sets are partitioned into 8:2 train-test splits per task. All models are trained for 30 epochs using the Adam optimizer with an initial learning rate of $10^{-3}$, employing MSE loss with gradient clipping (max_norm=1.0) and dropout regularization ($p = 0.3$). All experiments are repeated three times and take the mean value. All results represent mean values obtained from five independent replicate experiments.

### A.6   GAUSSIAN MODELING OF ACTIVATION DISTRIBUTIONS

Firstly, we briefly review the correspondence between single-layer neural networks and Guassian Processes similar to Matthews et al. (2018). As suggested, we employ task-specific linear encoders to transform each task's input into a representation of dimension $H$,

$$\mathbf{e}_t^{(k)} = \mathbf{W}_{\text{enc}}^{(k)} \mathbf{x}_t^{(k)} + \mathbf{b}_{\text{enc}}^{(k)}. \tag{53}$$

Here we emphasize the dependence on input $x$. Assume the weight and bias parameters to be i.i.d generated. Since $\mathbf{e}_t^{(k)}$ is a sum of i.i.d terms, from the Central Limit Theorem it follows that as $H \to \infty$, $\mathbf{e}_t^{(k)}$ will be Guassian distributed. Similarly, we have the following conclusion:

**Lemma 3** (Gaussianity of Activations). *Consider a neural network hidden layer with $n$ units. For task $k$, the hidden layer activation $h^{(k)} \in \mathbb{R}^n$ satisfies:*

$$h_i^{(k)} = \sum_{j=1}^{n} w_{ij} x_j^{(k)} + b_i. \tag{54}$$

*As $n \to \infty$ and weights $w_{ij}$ are independent and identically distributed, $h^{(k)}$ converges to a Gaussian distribution $\mathcal{N}(\mu^{(k)}, \Sigma^{(k)})$Neal (2012).*

*Proof.* Consider the standardized random variable:

$$Z_n = \frac{1}{\sqrt{n}} \sum_{j=1}^{n} (w_{ij} x_j^{(k)} - \mathbb{E}[w_{ij} x_j^{(k)}]). \tag{55}$$

By the Lindeberg-Feller Central Limit TheoremNeal (2012), when the Lindeberg condition is satisfied:

$$\lim_{n \to \infty} \frac{1}{s_n^2} \sum_{j=1}^{n} \mathbb{E}[(w_{ij} x_j^{(k)})^2 \mathbf{1}_{|w_{ij} x_j^{(k)}| > \epsilon s_n}] = 0, \tag{56}$$

where $s_n^2 = \sum_{j=1}^{n} \text{Var}(w_{ij} x_j^{(k)})$, then $Z_n \xrightarrow{d} \mathcal{N}(0,1)$.

In practical neural network settings, weights are typically initialized using Xavier or He initialization schemes, ensuring bounded variance and thus satisfying the Lindeberg condition. Therefore, $h_i^{(k)} \sim \mathcal{N}(\mu_i^{(k)}, \sigma_i^{(k)2})$, and collectively $h^{(k)} \sim \mathcal{N}(\mu^{(k)}, \Sigma^{(k)})$. $\square$

The above lemma demonstrates that for sufficiently deep neural networks, under parameter initialization, the output of each layer can indeed be assumed to exhibit Gaussian distribution. This Gaussianity propagates from layer to layer. For a detailed exposition and proof, refer to Matthews et al. (2018).

### A.7 SENSITIVITY ANALYSIS

As networks with varying numbers of communities can exhibit identical modularity values, in this section, we primarily investigate the impact of changing the number of communities on the model's stability while keeping the network's modularity constant.

Compared to the fluctuations induced by changes in $Q$, the variations in $CV$ values resulting from changes in the number of communities exhibit a significantly smaller average value (Fig. 4): For instance, the average $CV$ value computed across six tasks for different numbers of communities is 0.185, whereas the average $CV$ value attributable to $Q$ fluctuations is 0.426. Furthermore, on most tasks, the baseline $CV$ value substantially exceeds the change in $CV$ value induced by altering the number of communities; for example, on the Hénon Map, the $CV$ change due to variations in $C$ is 0.190, but the change due to $Q$ is 0.345. This suggests that, relative to the alterations in task accuracy caused by changes in $Q$, the variations resulting from modifying the number of communities are exceedingly minor. The sole exception occurs on the Lorenz system, where changes in the number of communities lead to more pronounced fluctuations in accuracy. This may be attributed to the more complex dynamical behavior of the Lorenz system, making it difficult to generate intricate dynamics when the number of communities is insufficient.

## B USE OF LLM

We acknowledge the use of large language models for text refinement and language polishing purposes during the preparation of this manuscript.

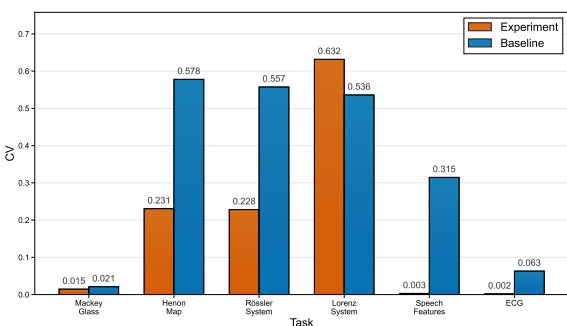

Figure 4: Performance stability comparison across prediction tasks, which compares varying community numbers $C$ (Experiment, orange) versus varying modularity $Q$ with fixed community numbers (Baseline, blue).

