# OpenReview forum: "Neuronal Network Modularity Determines Multi-Task Learning Capability"
_ICLR.cc/2026/Conference — Submitted to ICLR 2026_

### Official Review · Reviewer_TanJ · 2025-10-17

**Soundness:** 2
**Presentation:** 2
**Contribution:** 2
**Rating:** 2
**Confidence:** 3

**Summary:**

This paper argues that highly modular networks have limited capacities, using both theoretical analysis (an information-theoretic framework) and numerical experiments, although the common sense in neuroscience is that biological networks are highly modular. For multi-task setup, authors find that a certain level of modularity is desirable, while modularity does not seem to affect single tasks.

**Strengths:**

* This paper touches on an interesting topic -- how network modularity affects task performance
* The paper is written clearly and easy to follow

**Weaknesses:**

* The experimental part and the theoretical part are not seamlessly integrated. The theoretical part does not provide more insight into the experimental part. For example, can the theory predict the optimal Q = 0.4 for addition?
* I'm a bit confused by the theoretical part. What is the motivating question for developing such a theory? For example, why do we care about cross-module information flow, and how is the theory connected to practice?  The theory also shows that "too modular structure leads to limited capability", but to me, I think the more interesting question is the other side, "why do non-modular networks perform worse than intermediately modular networks?"
* Some presentation format can be improved, see Questions.
* In general, I think the paper presents several interesting results, but it does not have a consistent storyline.

**Questions:**

* why shows $\mu_{\rm norm}$ instead of un-normalized MSE? $\mu_{\rm norm}$ may exaggerate the performance difference.
* What is the addition task? It is used before making a definition.
* In Eq. 12, $\Sigma$ is not defined.
* citation format: in most places \cite => \citep, and in line 65-66, Yang et al. are displayed twice.
* Can you provide theoretical insight into why a certain level of modularity helps multi-task learning?

---

### Official Review · Reviewer_HRxc · 2025-10-23

**Soundness:** 2
**Presentation:** 2
**Contribution:** 1
**Rating:** 2
**Confidence:** 3

**Summary:**

This paper studies modular connectivity motifs in neural networks. The authors:
1. analyse connectomes to extract a quantification of their modularity
2. train neural networks  to perform a set of tasks, either in isolation or combination, and examine the impact of more or less modular connectivity.
3. derive results relating the information flow through the network to its connectivity strengths under gaussian assumptions

They conclude that:
1. There is an optimal modularity level for multi-task learning, something not shown by single-task learning
2. Show a quadratic dependence between cross-module mutual information and connection strengths

**Strengths:**

- It was clear what the authors had done, and the information required to understand their work was presented cleanly and logically.
- The authors present interesting results on the non-monotonicity of multi-task performance with respect to modularity, and performed the important test to show that the same results did not arise in single-task learning.
- The quadratic scaling of mutual information with norm of connectivity seems to be a novel result.

**Weaknesses:**

My major concern is that this felt like a set of relatively unrelated, minorly interesting results, bundled into a paper, muddying any overall contribution. What has this taught us about modularity and its role in neural networks?

First, the connectomic analysis was a footnote whose role for the rest of the paper was unclear, simply demonstrating that larger animals have more modularity according to their metric. Then the analytic results tell us that the more connected two modules, the higher their mutual information, a very natural conclusion that would have been anyone's guess without doing the maths. The only potentially specific prediction is the quadratic scaling of mutual information with connection strength, but this is not tested in real networks, nor is the value of this specific relationship clear, in which case, why worry about all the mathematics?

I thought the strongest part of the paper was the multi-task network, which shows an interesting non-monotonic relationship between performance and modularity. There seem to be a few potential claims that the authors could (and do) make from this result. First, that specific intermediate levels of modularity are useful for multi-task learning explaining their emergence in biology [a claim the authors do make], but to make this claim more than one example should really be given. Second, to generate understanding, they could have explored why this network performs well at multi-task learning. For example, does each task gets its own modules? Or are there shared motifs within the tasks that get assigned to certain modules? No analysis of this type is presented, limiting the amount we can learn. Third, they could argue this is a useful inductive bias to build into networks for applications [a claim the authors do make], but that would have required more evidence again than the single multi-task setting.

As such, this result seems interesting, but seems like it needs more work substantiating and clarifying to get into a venue like ICLR. Future directions that could be pursued are partially outlined in the paragraph above, but revolve around either (i) robustly demonstrating this phenomenon in a sufficiently broad set of multi-task settings to motivate the idea it is a general link between modularity and multi-task learning (ii) doing some mechanistic uncovering of why this link occurs or (iii) showing its use in an application direction. All of these comprise significant research directions that I would not expect a rebuttal to cover, as such I'm afraid I must recommend rejection and resubmission at a later date.

Minor
- I think each datapoint was the result of a set of simulations, could you plot the errorbars so that we have some idea of what a meaningful amount of variation looks like?

**Questions:**

- What does ER stand for in Table 1, Erdos-Renyi? Maybe I missed it, if not, should be included.
- Somewhere in the paper we should see loss values, not just normalised, to see how well the network is performing on the task.
- Are all tasks trained concurrently in the multi-task setting?

---

### Official Review · Reviewer_nury · 2025-10-29

**Soundness:** 2
**Presentation:** 1
**Contribution:** 2
**Rating:** 2
**Confidence:** 4

**Summary:**

This paper investigates the relationship between network modularity and multi-task learning performance from an information-theoretic perspective. Motivated by the modular organization observed in biological connectomes, the authors first quantify modularity across four species-specific neural networks. They then develop a multi-task learning framework using recurrent neural networks trained on six time-series tasks. The RNN architecture is designed such that connectivity between hidden states across time steps is governed by a modular binary adjacency matrix, where the degree of modularity can be systematically varied. Specifically, modularity is controlled by adjusting the ratio of intra-module to inter-module connection probabilities, quantified using the Newman-Girvan modularity metric (Q).

The results reveal a non-monotonic relationship between modularity and performance: as Q increases, loss initially decreases before increasing again at extreme modularity levels. Notably, single-task learning exhibits no systematic pattern with varying Q, suggesting that modularity's effects are specific to multi-task scenarios. The authors hypothesize that extreme modularity causes performance degradation due to restricted cross-module information flow. They provide theoretical support by demonstrating that mutual information between modules exhibits quadratic dependence on inter-module connection strength for weakly connected modules, offering a mathematical explanation for the observed performance degradation.

**Strengths:**

- The research addresses a crucial area with well-founded motivations. The authors' independent analysis of biological networks adds value, though the modular nature of these networks is well-established.

- The experimental methodology is sound, employing established network science tools, including appropriate modularity metrics and systematic methods for generating modular adjacency matrices with variable degree of modularity.

**Weaknesses:**

- The experimental scope and result analysis lacks sufficient depth, making it challenging to understand the significance of the findings. The study would benefit from experiments incorporating a broader range of tasks and examining varying number of task, rather than focusing solely on the extreme scenarios of 1 task vs. all tasks.

- The homogeneous nature of the tasks, while suitable for recurrent networks, raises questions about the generalizability to other network architectures and task types.

- The paper is often difficult to comprehend, with critical concepts and details either omitted or referred to in the appendix. this includes module definition in the context of network and the tasks, the task and the training details. These details should be integrated into the main text, while maintaining detailed proofs in the appendix with their logical progression outlined in the main body. Additionally, the theoretical section's assumptions about module constitution lack clarity due to lack of details.

**Questions:**

- How does this work define modules? The transition between experimental and theoretical frameworks appears to employ different module conceptualizations. This distinction requires clearer explanation.

- While the mathematical analysis is sound, what mechanisms link reduced inter-module information flow to decreased performance? Could this suggest that learned representations aren't being shared effectively, essentially forcing highly modular systems to operate as isolated, smaller networks?

- Can you provide an architectural diagram for better understanding of the architecture and the modules ?

- I also encourage the authors to expand the related works to include other modularity, generalization and multi-task learning studies to place the paper well in the literature. Recent literature and references are completely missing from the current version.

---

### Official Review · Reviewer_yooC · 2025-10-31

**Soundness:** 2
**Presentation:** 3
**Contribution:** 2
**Rating:** 2
**Confidence:** 4

**Summary:**

The paper investigates how network modularity influences performance in multitask learning.

Using a fixed recurrent neural network architecture with controlled modular connectivity, the authors vary modularity while keeping the number of nodes and total connections fixed.

Experiments across several time series tasks show a non-monotonic trend, with intermediate modularity leading to the best multitask performance, while single-task training shows no consistent dependence.

A theoretical analysis is provided to argue that excessive modular separation limits inter-module information flow, though this analysis is largely qualitative.

The work aims to draw connections between modularity in artificial networks and biological neural systems, suggesting that moderate modular organization may support flexible task learning.

**Strengths:**

The paper addresses an interesting question about how modular network structure influences multitask learning performance.

Results are intriguing and suggest a consistent, non-monotonic relationship between modularity and multitask performance. The single-task vs. multitask comparison is a useful control.

The work connects to ideas in neuroscience and offers an intuitively plausible explanation for the observed trends.

**Weaknesses:**

The “sweet spot” Q≈0.4  is interesting. However, there is no discussion of a proposed explanation for why there might be a non-monotonic relation between modularity and multi-task performance.

Morevoer, no comparison is given with the modularity values reported in Table 1, tying the results back to the biology. Table 1 shows that different animals can have quite different values for Q, and the ones with value closest to 0.4 are the two larvae. The authors say that all animals have "significantly elevated" modularity -- doesn't that conflict with the paper's conclusion that high modularity may be detrimental: there, the authors say that biological systems evolved "moderate modularity levels".

The conclusions are limited because all experiments are conducted on a single arbitrary recurrent architecture and one sparsity parameterization. There is no systematic comparison across model families (e.g., LSTM, GRU, Transformer) or task domains beyond the selected six time-series datasets (e.g., what happens as the number of tasks increase?). Thus, it's unclear whether the reported non-monotonic relationship generalizes to other architectures or modalities/datasets.

In addition, the quantitative results appear to be averaged over only three random initializations (or five? this number is reported inconsistently in Appendix A.5), which provides too small a sample to assess variability or statistical significance. Moreover, no hypothesis tests/intervals are presented; normalized MSE curves lack error bars in Fig. 1.

The authors should also report per-task errors, not just min–max-scaled aggregates.
Moreover, it is misleading to use min-max normalization for $\mu_{norm}$. One problem, e.g., is that it obscures absolute improvements (e.g., a 1 % vs. 10 % reduction could both map to a “0.1 drop” after normalization); it can also distort variance if any outlier setting yields an extreme MSE, since min–max compresses all other values.
An MSE of zero should be absolute: it has the same meaning for any task, i.e. the model perfectly fits the data. A better normalization could be to simply divide the error of each task by the standard deviation of the predicted variable (as in Normalized RMSE); another standard choice is to compute the relative absolute error (RAE).

Section 4 seems rather disconnected from the rest of the paper. Although it aims to provide a mechanistic justification for why too much modularity may harm multi-task learning, the theory predicts a monotonic increase of inter-module information with coupling strength, not an optimal intermediate point. The “non-monotonic” pattern from Fig. 1 arises only empirically, so Section 4 doesn’t actually explain the observed optimum. Moreover, this section reads more like an appendix-style theoretical note than an argument integrated with the biologically-motivated study up to that point. I believe moving it to an appendix would improve coherence, which would leave room for a more thorough analysis of the empirical results, testing more models and variations on the set of tasks.

Finally, the authors claim that their results “explain the evolutionary conservation of modular brain architecture from \textit{C. elegans} to humans”. This claim is exaggerated: the paper’s experiments are conducted on synthetic recurrent networks trained on abstract multitask regression tasks, without evolutionary modeling or behavioral data. At best, the results offer a useful analogy, not an explanation, for modularity in evolution. I suggest rephrasing it to something like "our results are consistent with the hypothesis that modularity can support multitask flexibility observed in biological systems”.


__Minor points__:

Parsing equations 2--5 with so many quantities and indices requires some effort (lines 131--140 are not very helpful). Since all the results in the paper depend on this particular model, the authors could facilitate that understanding by adding a small flow-chart diagram to illustrate the overall architecture (including the decoding head (eqs. 6--10) as well).

There are many typos in the paper (please use a spell-checker), and authors should enter a space before the `\citep` command in LaTeX.

**Questions:**

- How many independent runs per point in Fig. 1? Please resolve the “3 vs. 5” inconsistency (Appx. A.5) and add error bars/significance tests.

- Results fix edge count and hidden units across all configurations. Does the optimal Q has any significant shift with different densities (e.g., 1k, 4k edges) or hidden layer sizes?

- What's an "ER baseline" for modularity (Table 1)?

---

### Meta-Review · Area_Chair_p8dk · 2025-12-09

**Summary:**

The paper studies the role of modularity in neural network multitasking. Reviewers agreed that the topic is interesting and the results are intriguing, but all raised concerns about the limited experimental scope and about the lack of consolidation and discussion of the results into a clear picture. The authors did not participate in the rebuttal phase. The paper is therefore recommended for rejection in the current submission and revision and resubmission.

**Reviewer Concerns:**

N/A (no author rebuttal submitted).

**Reviewer Scores:**

No author rebuttal was submitted, nor would the reviewers have changed their mind in internal discussions since they gave the same score and raised similar concerns.

---

### Decision · Program_Chairs · 2026-01-26

Reject